# Genome wide association study of clinical duration and age at onset of sporadic CJD

Holger Hummerich[1], Helen Speedy[1], Tracy Campbell[1], Lee Darwent[1], Elizabeth Hill[1], Steven Collins[2], Christiane Stehmann[2], Gabor G. Kovacs[3,4,5], Michael D. Geschwind[6], Karl Frontzek[7], Herbert Budka[5], Ellen Gelpi[5], Adriano Aguzzi[7], Sven J. van der Lee[8,9,10], Cornelia M. van Duijn[11,12], Pawel P. Liberski[13], Miguel Calero[14], Pascual Sanchez-Juan[15], Elodie Bouaziz-Amar[16], Jean-Louis Laplanche[16], Stéphane Haïk[17,18], Jean-Phillipe Brandel[17,18], Angela Mammana[19], Sabina Capellari[19,20], Anna Poleggi[21], Anna Ladogana[21], Maurizio Pocchiari[21], Saima Zafar[22,23], Stephanie Booth[24], Gerard H. Jansen[25], Aušrinė Areškevičiūtė[26], Eva Løbner Lund[26,27], Katie Glisic[28], Piero Parchi[19,20], Peter Hermann[22,29], Inga Zerr[22,29], Brian S. Appleby[28], Jiri Safar[30], Pierluigi Gambetti[30], John Collinge[1], Simon Mead[1]*

1 MRC Prion Unit at University College London (UCL), Institute of Prion Diseases, UCL, London, United Kingdom, 2 Australian National Creutzfeldt-Jakob Disease Registry, The Florey, Department of Medicine (RMH), The University of Melbourne, Victoria, Australia, 3 Department of Laboratory Medicine and Pathobiology and Tanz Centre for Research in Neurodegenerative Disease, University of Toronto, Ontario, Toronto, Canada, 4 Laboratory Medicine Program & Krembil Brain Institute, University Health Network, Toronto, Ontario, Canada, 5 Division of Neuropathology and Neurochemistry, Department of Neurology, Medical University of Vienna and Austrian Reference Center for Human Prion Diseases (ÖRPE), Vienna, Austria, 6 UCSF Memory and Aging Center, Department of Neurology, University of California, San Francisco, California, United States of America, 7 Institute of Neuropathology, University of Zürich, Zürich, Switzerland, 8 Section Genomics of Neurodegenerative Diseases and Aging, Department of Clinical Genetics, Vrije Universiteit Amsterdam, Amsterdam UMC, Amsterdam, The Netherlands, 9 Delft Bioinformatics Lab, Delft University of Technology, Delft, The Netherlands, 10 Amsterdam Neuroscience, Neurodegeneration, Amsterdam, The Netherlands, 11 Nuffield Department of Population Health, University of Oxford, Oxford, United Kingdom, 12 Department of Epidemiology, Erasmus Medical Centre, Rotterdam, The Netherlands, 13 Department of Molecular Pathology and Neuropathology, Medical University of Lodz, Lodz, Poland, 14 Chronic Disease Programme (UFIEC-CROSADIS) and Network Center for Biomedical Research in Neurodegenerative Diseases (CIBERNED), Instituto de Salud Carlos III, Madrid, Spain, 15 Alzheimer's Centre Reina Sofia-CIEN Foundation-ISCIII, Research Platforms, Madrid, Spain, 16 Department of Biochemistry and Molecular Biology, Lariboisière Hospital, GHU AP-HP Nord, University of Paris Cité, Paris, France, 17 Paris Brain Institute (Institut du Cerveau, ICM), INSERM, CNRS, Assistance Publique-Hôpitaux de Paris (AP-HP), Sorbonne Université, Paris, France, 18 Assistance Publique-Hôpitaux de Paris (AP-HP), Cellule Nationale de Référence des Maladies de Creutzfeldt-Jakob, Groupe Hospitalier Pitié-Salpêtrière, Paris, France, 19 IRCCS, Istituto delle Scienze Neurologiche di Bologna, Bologna, Italy, 20 Department of Biomedical and Neuromotor Sciences, University of Bologna, Bologna, Italy, 21 Department of Neuroscience, Istituto Superiore di Sanità, Rome, Italy, 22 Department of Neurology, Clinical Dementia Center and National Reference Center for CJD Surveillance, University Medical School, Göttingen, Germany, 23 Biomedical Engineering and Sciences Department, School of Mechanical and Manufacturing Engineering, National University of Sciences and Technology, Islamabad, Pakistan, 24 Prion Disease Program, National Microbiology Laboratory, Public Health Agency of Canada, Winnipeg, Canada, 25 Department of Pathology and Laboratory Medicine, University of Ottawa, Ottawa, Canada, 26 Danish Reference Center for Prion Diseases, Department of Pathology, Copenhagen University Hospital, Rigshospitalet, Copenhagen, Denmark, 27 Department of Clinical Medicine, University of Copenhagen, Copenhagen, Denmark, 28 National Prion Disease Pathology Surveillance Center, Case Western Reserve University, Cleveland, OH, United States of America, 29 German Center for Neurodegenerative Diseases (DZNE), Göttingen, Germany, 30 Departments of Pathology, Case Western Reserve University School of Medicine, Cleveland, OH, United States of America

* s.mead@prion.ucl.ac.uk

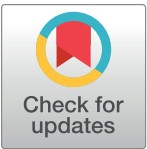

**Data Availability Statement:** The data used in this study have been deposited in the public database GWAS Catalogue (https://www.ebi.ac.uk/gwas/; accession number GCP000963).

**Funding:** This work was supported by the MRC (UK) core grant to the MRC Prion Unit at UCL (code MC_UU_00024/1). Several authors at UCL/UCLH receive funding from the Department of Health's NIHR Biomedical Research Centres funding scheme. Some of this work was supported by the Department of Health funded National Prion Monitoring Cohort study. Funding for the collection of Polish samples for study was partially provided by the EU joint programme JPND and Medical University of Lodz. The Italian national surveillance of Creutzfeldt-Jakob disease and related disorders is partially supported by the Ministero della Salute, Italy. The German National Reference Centre for TSE is funded by grants from the Robert-Koch-Institute. The Dutch National Prion Disease Registry is funded by the National Institute for Public Health and the Environment (RIVM), which is part from the Ministry for Health, Welfare and Sports, The Netherlands. PS-J was supported by Instituto de Salud Carlos III [Fondo de Investigación Sanitaria, PI16/01652] Accion Estrategica en Salud integrated in the Spanish National I+D+i Plan and financed by Instituto de Salud Carlos III (ISCIII) – Subdireccion General de Evaluacion and the Fondo Europeo de Desarrollo Regional (FEDER – "Una Manera de Hacer Europa"). The French National Surveillance Network for Creutzfeldt-Jakob disease is supported by Santé Publique France. MDG (UCSF) receives research support from the NIH/NIA (grant R01 AG031189, R56AG055619, R01AG062562) and the Michael J. Homer Family Fund. The National Prion Disease Pathology Surveillance Center in the U.S. is funded by the Centers for Disease Control and Prevention (NU38CK000486). The Austrian Reference Center for Human Prion diseases (ÖRPE) is supported by the Austrian Ministy of Health – Bundesministerium für Soziales, Gesundheit, Pflege und Konsumentenschutz. The ANCJDR is supported through funding from the Commonwealth Department of Health and Aged Care.

**Competing interests:** Stéphane Haik reports grants from Santé Publique France, during the conduct of the study; grants from LFB Biomedicaments, grants from Institut de Recherche Servier, grants from MedDay Pharmaceuticals, outside the submitted work; In addition, Stéphane Haik has a patent Method for treating prion diseases (PCT/EP2019/070457) pending. Brian Appleby has received funding from CDC, NIH, CJD Foundation, Alector, and Ionis. He has served as a consultant for Ionis, Sangamo, and Gate Biosciences. He has received royalties from Wolter Kluwers. Karl Fronztek reports grants from Ono Pharmaceuticals outside the submitted work. Simon Mead reports

## Abstract

Human prion diseases are rare, transmissible and often rapidly progressive dementias. The most common type, sporadic Creutzfeldt-Jakob disease (sCJD), is highly variable in clinical duration and age at onset. Genetic determinants of late onset or slower progression might suggest new targets for research and therapeutics. We assembled and array genotyped sCJD cases diagnosed in life or at autopsy. Clinical duration (median:4, interquartile range (IQR):2.5–9 (months)) was available in 3,773 and age at onset (median:67, IQR:61–73 (years)) in 3,767 cases. Phenotypes were successfully transformed to approximate normal distributions allowing genome-wide analysis without statistical inflation. 53 SNPs achieved genome-wide significance for the clinical duration phenotype; all of which were located at chromosome 20 (top SNP rs1799990, pvalue = $3.45 \times 10^{-36}$, beta = 0.34 for an additive model; rs1799990, pvalue = $9.92 \times 10^{-67}$, beta = 0.84 for a heterozygous model). Fine mapping, conditional and expression analysis suggests that the well-known non-synonymous variant at codon 129 is the obvious outstanding genome-wide determinant of clinical duration. Pathway analysis and suggestive loci are described. No genome-wide significant SNP determinants of age at onset were found, but the *HS6ST3* gene was significant (pvalue = $1.93 \times 10^{-6}$) in a gene-based test. We found no evidence of genome-wide genetic correlation between case-control (disease risk factors) and case-only (determinants of phenotypes) studies. Relative to other common genetic variants, *PRNP* codon 129 is by far the outstanding modifier of CJD survival suggesting only modest or rare variant effects at other genetic loci.

## Introduction

Human prion diseases are rare and often rapidly progressive dementia disorders with no known treatments that slow the disease process. The most common type, sporadic Creutzfeldt-Jakob disease (sCJD), occurs at a relatively uniform annual incidence of 1-2/million, equating to a lifetime risk of approximately 1:5000 [1]. The clinical presentation and progression of the disorder is remarkably variable both in terms of the initial symptoms and signs, age at onset and clinical duration [2–4]. Patients typically present in late middle or old age but have been reported in adolescence and early adulthood, and at the extremes of old age [5–7]. The median clinical duration is usually reported as five months with a range of only a few weeks to several years [2]. Ability to estimate the likely clinical duration could help with timely decisions about care [8].

Prions are proteinaceous pathogens formed of host prion protein (PrP) which cause mammalian prion diseases like bovine spongiform encephalopathy, sheep scrapie, chronic wasting disease of cervids, and the human disorders [9]. The recently determined structures of mouse and hamster prions reveals assemblies of PrP in a parallel in-register beta sheet structure with two domains [10, 11], in marked contrast to the predominant alpha-helices of normal cellular PrP [12]. Prions are thought to replicate by a process of binding of normal cellular PrP, conformational change and subsequently aggregate fission. In several model systems, incubation time of prion disease is influenced by PrP gene expression, primary sequence and polymorphisms, as well as prion strains [13], thought to be conferred by structural variation of the pathogen [14]. Experiments using animal or cellular model systems have led to proposals of

grants from Medical Research Council (UK) and grants from National Institute of Health Research's Biomedical Research Centre at University College London Hospitals NHS Foundation Trust during the conduct of the study. Gabor G Kovacs reports personal fees from Biogen, outside the submitted work. John Collinge reports grants from Medical Research Council, grants from NIHR UCLH Biomedical Research Centre, during the conduct of the study; and is a Director and shareholder of D-Gen Limited, an academic spinout in the field of prion disease diagnostics, decontamination and therapeutics. Inga Zerr reports grants from the Bundesministerium für Gesundheit via Robert Koch institute, JPND and personal fees (not related to the content of the manuscript) from Ferring Pharmaceuticals and IONIS, speaking honoraria for medical lectures from Lilly, Biogen, Medfora, DGLN (German Society for cerebrospinal fluid diagnostics in Neurology). Maurizio Pocchiari reports personal fees from Ferring Pharmaceuticals, personal fees from CNCCS (Collection of National Chemical Compounds and Screening Center), non-financial support from Fondazione Cellule Staminali, outside the submitted work. Michael D Geschwind has consulted for3D Communications, Adept Field Consulting, Advanced Medical Inc., Best Doctors Inc., Second Opinion Inc., Gerson Lehrman Group Inc., Guidepoint Global LLC, InThought Consulting Inc., Market Plus, Trinity Partners LLC, Biohaven Pharmaceuticals, Quest Diagnostics and various medical-legal consulting. He has received speaking honoraria for various medical center lectures and from Oakstone publishing. He has received past research support from Alliance Biosecure, CurePSP, the Tau Consortium, and Quest Diagnostics. Michael D Geschwind serves on the board of directors for San Francisco Bay Area Physicians for Social Responsibility and on the editorial board of Dementia & Neuropsychologia.

several possible non-PrP mechanisms of toxicity in prion diseases, involving PrP binding partners on the cell surface and downstream intracellular changes [15–17]; however, their relevance to the human diseases is yet to be determined.

Human epidemiological and genetic studies have identified factors that associate with survival time in sCJD [2, 8, 18, 19], including demographic factors, prion protein genotype, molecular strain typing of protease-resistant prion protein by Western blot analysis, and a range of biofluid, tissue, imaging, and neurophysiological biomarkers [20]. Many biomarkers simply measure the rate or extent of neuronal injury, loss, or dysfunction, or immune cell or glial responses, whereas genetic associations are implicitly causal of modified clinical phenotypes. In this study, we sought to determine the effects of genome-wide common genetic variation on key clinical phenotypes of sCJD, to develop evidence of modifiers relevant to human prion diseases that might benefit understanding of disease processes and generate new ideas for therapeutics.

## Materials and methods

### Diagnosis and clinical phenotypes

Details of the contributing sites and diagnostic criteria were given in a previous publication [19]. In short, all patient participants were deceased and gained a diagnosis in life of probable CJD or definite CJD after a post-mortem examination (using contemporary epidemiological criteria which changed over the recruitment period 1990–2019). "Probable CJD" is an epidemiological term that now equates to an almost certain diagnosis of CJD post-mortem (e.g. [21]). Age at clinical onset was given to the nearest month. Clinical duration was based on the examining physician's impression of the date of onset of the first symptom that subsequently was thought to be a component of the disease syndrome until death in months.

Samples used in this study were obtained over several decades and the data were accessed from January 2023 until now.

### Genotyping and quality control

In addition to 4110 samples previously reported, genotyped on an Illumina OmniExpress array [19], 819 new samples were genotyped using Illumina's Global Screening Array. Standard sample and genotyping quality control was performed using PLINK v1.90b3v, which generated 6,308,901 autosomal SNPs of high quality. Samples with a call rate below 98% and population outliers identified via multidimensional scaling were removed. Additionally, related samples (Pi_Hat > 0.1875) were discarded. Only autosomal SNPs with a genotyping rate of >99%, a minor allele frequency $\geq 0.01$ and SNPs not deviating from the Hardy-Weinberg equilibrium ($P > 10^{-4}$) were retained. SNPs of A/T or G/C transversion or those which showed deviation from heterozygosity mean (±3 SD) were excluded. To ensure consistency with the Michigan Imputation Server pipeline the target VCF files were checked against the 1000 Genomes Project reference panel (https://faculty.washington.edu/browning/conform-gt.html/). Genotypes were imputed using the Michigan Imputation Server (using Minimac4 assuming a mixed population, HRC r1.1 2016 (Haplotype Reference Consortium) as reference panel and Eagle 2.4 for phasing) [22]. A post-imputation QC analysis was carried out and SNPs with an $r^2$ threshold lower than 0.3 (removing 70% of poorly imputed SNPs) were excluded.

## Statistical analysis

SNPTEST (v2.5.2) was used to perform association and conditional analysis with an additive and heterozygous logistic regression model, using sex, contributing site and 10 population covariates generated with PLINK (v1.90b3v; www.cog-genomics.org/plink/1.9/). Genetic correlation between this (using duration as phenotype) and the previously conducted sCJD case-control study [19] was performed using LDSC [23], a software tool for linkage disequilibrium (LD) score and heritability estimation using summary statistics. Meta-analysis was performed using METAL combining the previously published GWAS case-control data [19] and the case-only data described here using summary test statistics as input (6,314,883 SNPs in the union list) and adopting the sample-based approach by combining z-scores across samples in a weighted sum proportional to study sample sizes. FUMA [24], using an integrated Magma gene-based and gene-set analysis on the GWAS summary data, was utilised to perform pathway analysis to identify genes and pathways associated with sCJD risk. FUMA also provides information about chromatin interaction, expression patterns and shared molecular functions between genes. MAGMA software was also utilised for gene-based / gene-set analysis [25]. Power analysis was performed using R functions taken from the Github site https://github.com/kaustubhad/gwas-power provided by Kaustubh Adhikari (UCL Division of Biosciences, University College London).

## Ethics

The research project has approval from the NHS Health Research authority (London—Harrow Research Ethics Committee, London, UK); the REC reference is 05/Q0505/113. Written informed consent has been obtained.

## Results

We performed the association analysis with 3773 (duration as phenotype; median:4.0, IQR:2.5–9 (months)) and 3767 (age at onset as phenotype; median:67, IQR:61–73 (years)). cases of probable or definite sCJD by contemporary diagnostic criteria either included in a previous paper from the collaborative group [19], or newly genotyped on Illumina's Global Screening Array (Table 1). All patients were deceased.

**Table 1. Number of samples used in the association test from 12 countries (duration / age) with interquartile range and median.**

| Country | N (duration) | N (age) | Median (duration) | IQR (duration) | Median (age) | IQR (age) |
|---|---|---|---|---|---|---|
| Australia | 22 | 22 | 2.05 | 1.87 | 67 | 12.5 |
| Austria | 44 | 44 | 4.5 | 5.87 | 72 | 7 |
| Canada | 133 | 133 | 4 | 5 | 67 | 14 |
| France | 95 | 95 | 4 | 4 | 68 | 13 |
| Germany | 798 | 792 | 6 | 8 | 66 | 12 |
| Italy | 554 | 554 | 5 | 7 | 67 | 13 |
| Netherlands | 126 | 126 | 3.94 | 4.02 | 66.5 | 13 |
| Poland | 42 | 42 | 3 | 2.88 | 63.5 | 9.25 |
| Spain | 74 | 74 | 3.45 | 4.25 | 69 | 13.75 |
| Switzerland | 35 | 35 | 2.69 | 2.91 | 70 | 13.5 |
| UK | 951 | 951 | 5 | 6.95 | 67 | 12 |
| USA | 899 | 899 | 3 | 5 | 67 | 13 |
| Total | 3773 | 3767 | | | | |

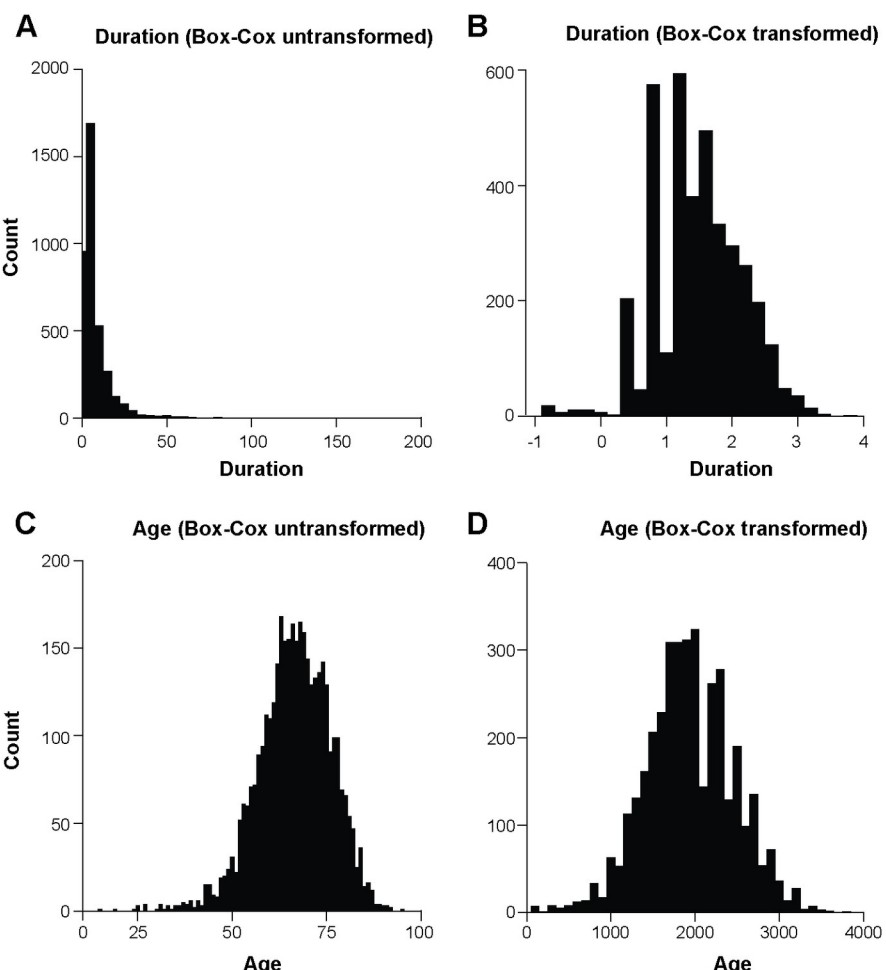

**Fig 1.** Histograms for phenotypes duration before (A) and after (B) Box-Cox transformation and age before (C) and after (D) Box-Cox transformation.

Genotype doses were imputed using the Michigan Imputation Server [22], resulting in 6,308,901 SNPs passing quality control.

The median age / duration for men was 67 years and 3.8 months respectively and 67 years and 4.0 months for women. Median clinical duration (2.0–6.0 months) and age at onset (63.5–72 years) varied by site, so this was included as a covariate in the analysis. Phenotypes were modelled as normally distributed quantitative traits following transformation using methods developed by Box and Cox [26] illustrated as histograms and QQ plots (Figs 1 and 2; S1 Fig). Association analysis omitting sex, age or country or any combination as covariates did not show any significant difference in terms of outcome. Principal components analysis was used to exclude cases with distinct ancestry (n = 54) and did not suggest any strong effects of ancestry on the outcomes of interest (S2 and S3 Figs).

Additive and heterozygous genetic models were run genome-wide in SNPTEST with sex, contributing site and genetic ancestry covariates (see Methods) without any statistical inflation (lambda = 1.000 / 1.000 for clinical duration / age) as illustrated with QQ plots in Fig 3 (duration phenotype) and Fig 4 (age phenotype). 53 SNPs achieved genome-wide significance ($P < 5 \times 10^{-8}$) for the clinical duration phenotype (additive model) (Fig 5 and S1 Table), all at the *PRNP* locus (top SNP rs1799990, pvalue = $3.45 \times 10^{-36}$, beta = 0.34 for additive model;

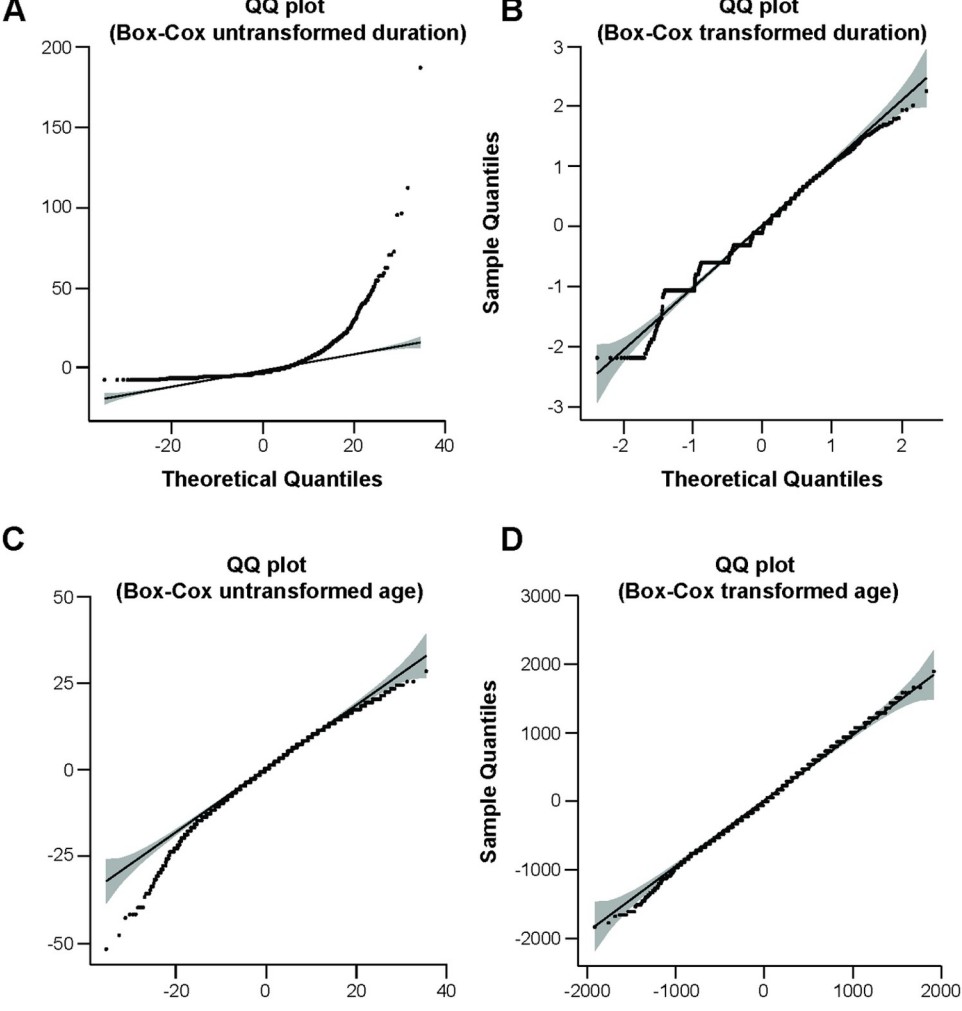

**Fig 2.** Quantile-Quantile plots for phenotypes duration before (A) and after (B) Box-Cox transformation and age before (C) and after (D) Box-Cox transformation.

rs1799990, pvalue = $9.92 \times 10^{-67}$, beta = 0.84 for heterozygous model, Figs 6 and 7). *PRNP* rs1799990 was the obvious outstanding genome-wide candidate determinant of clinical duration.

Of 68 cis-eQTL SNPs associated with *PRNP* expression in various brain tissues (obtained from GTEx); none were present in the list of 53 SNPs achieving genome-wide significance (duration phenotype). 50 of these eQTL SNPs for *PRNP* passed QC, all were P>0.001 (duration phenotype). No genome-wide significant SNPs remained after conditioning for rs1799990 codon 129 (Fig 8 and S4 Fig). There were 51 suggestive associated SNPs ($5 \times 10^{-8}$ > pvalue<$1 \times 10^{-5}$, including at regions near to *HDHD5* (chromosome 22), *FHIT* (chromosome 3) and *EREG* (chromosome 4) (S2 Table and S5–S7 Figs). There were no significant gene-based associations of clinical duration apart from *PRNP* (MAGMA and FUMA) (Tables 2 and 3).

Age-based analysis did not identify any genome-wide significant SNP associations (Fig 9). Two suggestive associations were identified on chromosome 15 near *NEDD4* and chromosome 13 near *UGGT2* (S8 and S9 Figs). Gene-based analysis for age at onset with MAGMA identified *HS6ST3* (pvalue = $1.93 \times 10^{-6}$), with similarly significant association detected using FUMA (S3 and S4 Tables).

## Q-Q plot (duration)

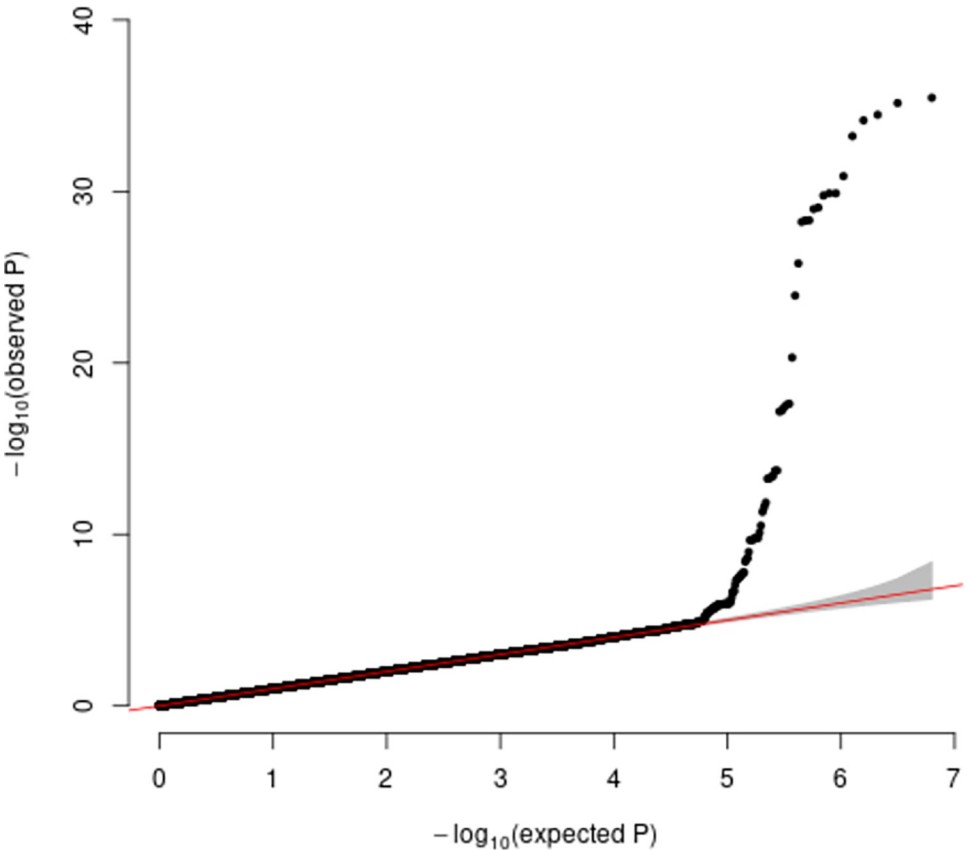

**Fig 3. Quantile-Quantile plot with duration as phenotype.**

Gene-set analysis for clinical duration using FUMA (including *PRNP* locus) identified binders of type-5 metabotropic glutamate receptors (GO Molecular Function ontology n = 1738, pvalue = 1.85 x $10^{-5}$) (Tables 4 and 5). Gene-set analysis for age at onset using MAGMA revealed intracellular oxygen homeostasis as a significant term (pvalue = 1.89 x $10^{-6}$) (S5 Table). Genetic correlation between clinical duration GWAS and the previously published case-control GWAS resulted in a non-significant genetic correlation of 0.1467 (pvalue = 0.79, 95% CI 0.92,1.21; S6 Table). Meta-analysis of the two GWAS (case-only and case-control) resulted in the same strong codon 129 effect as described above whilst removing the suggestive locus on chromosome 22 the *HDHD5* locus (S10 Fig).

We also calculated the power of the study based on 3773 samples and a genome-wide significance level of 5x$10^{-8}$ using the additive model with a range of effect sizes and minor allele frequencies. Plotting the most significant SNP (*PRNP*; rs1799990) and the lead SNPs of the suggestive association signals (*HDHD5*, rs4819962; *FHIT*, rs2366847; *EREG*, rs11727991) resulted in rs1799990 achieving full power and the three lead SNPs being borderline achieving a power value of ~0.7–0.8 (S11 Fig).

Interestingly, there was no evidence that the sCJD genetic susceptibility genes, *STX6* or *GAL3ST1*, which were identified in the previously published case-control study [19], modify clinical phenotypes. The identification of these genes in the case-control GWAS implicated

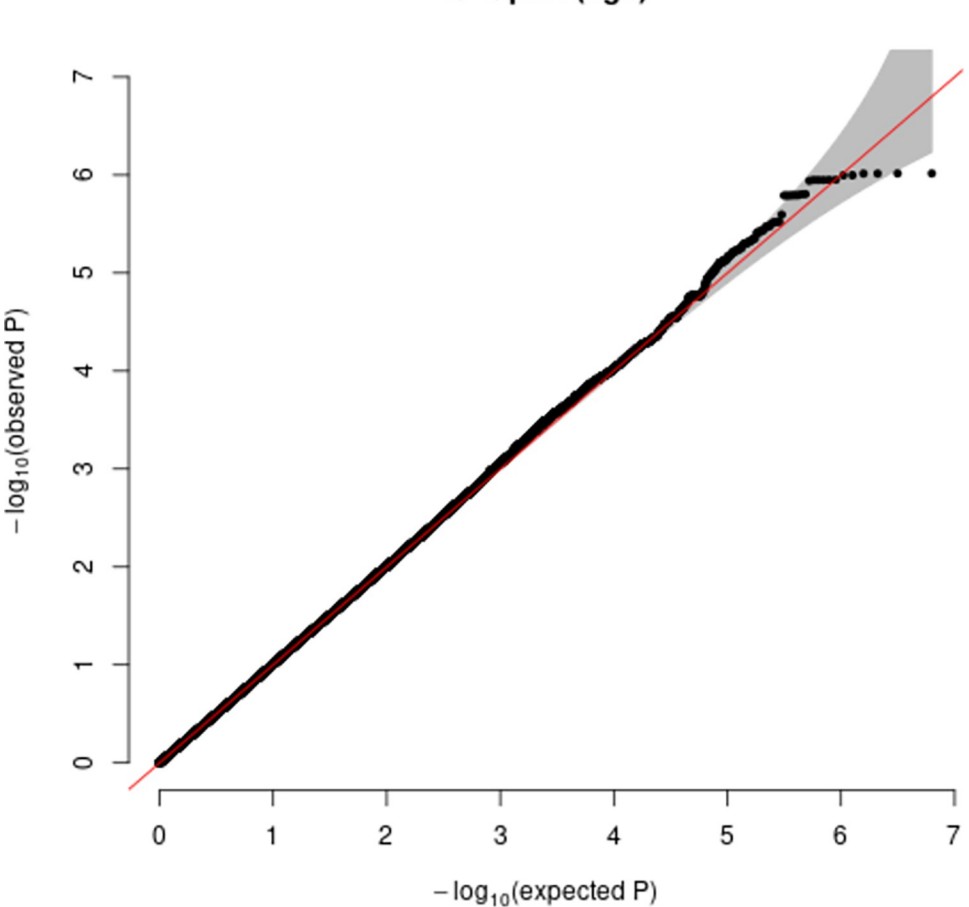

**Fig 4. Quantile-Quantile plot with age as phenotype.**

intracellular trafficking and sphingolipid metabolism respectively as causal disease mechanisms. To further investigate the roles of these pathways in disease phenotypes, we compiled a comprehensive, bespoke gene list including genes related to these pathways, which have been implicated in neurodegenerative diseases, and performed MAGMA analysis (S7 and S8 Tables). This highlighted *UGGT2*, a sphingolipid metabolism linked gene, to be associated with sCJD age of onset.

## Discussion

We describe the first well-powered GWAS for phenotypic traits in sporadic human prion disease. The only clearly identified risk locus was the *PRNP* gene itself, more specifically the well-known common variant at codon 129, for the clinical duration phenotype. Conditioning for the codon 129 polymorphism at this locus removed all evidence of association at the locus, implicating the coding sequence of *PRNP* and not PrP expression in controlling this phenotype. We found a number of suggestive risk loci with $P < 10^{-5}$, which should require additional genetic evidence before being considered further. Pathway analysis identified binders of type-5 metabotropic glutamate receptors, which are known to mediate the downstream effects of amyloid beta bound to prion protein, as a top hit for clinical duration [27, 28]. Importantly however, since this small gene set (n = 5) was non-significant after removing *PRNP*, these data

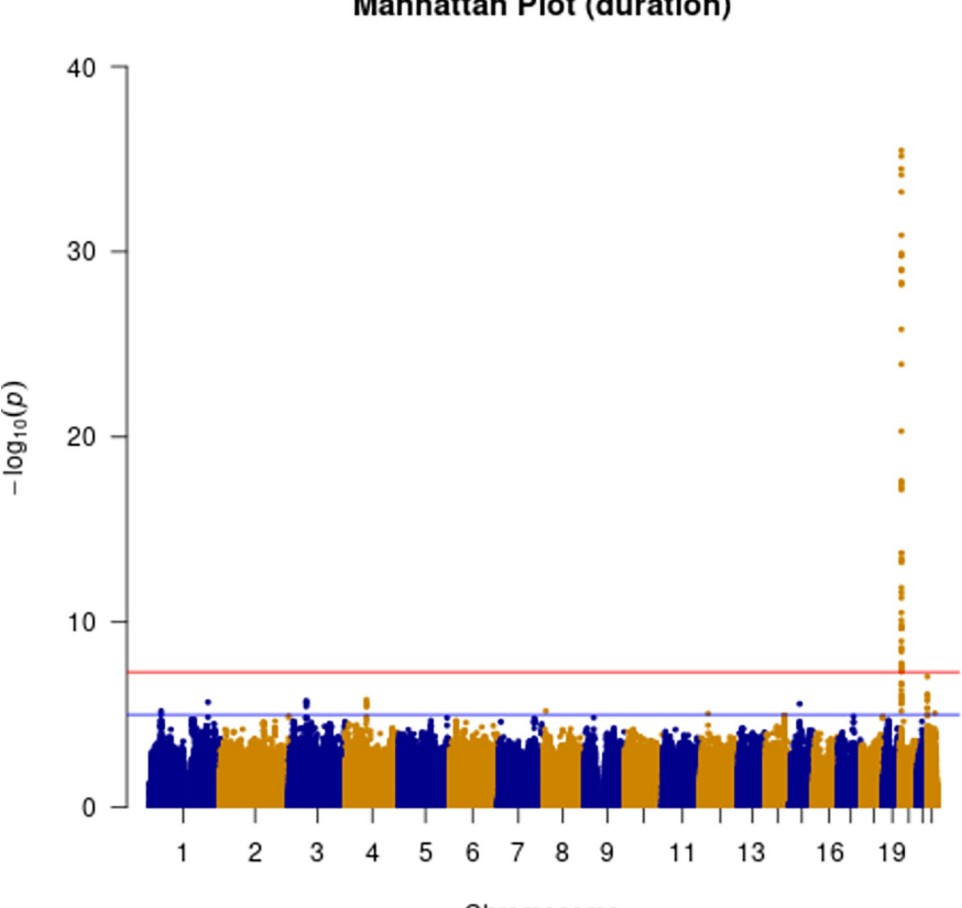

**Fig 5. Manhattan plot with clinical duration as phenotype.** (red line indicating genome-wide significance of $5\times10^{-8}$; blue line indicating suggestive genome-wide significance ($5\times10^{-8} > $ pvalue $< 1\times10^{-5}$)).

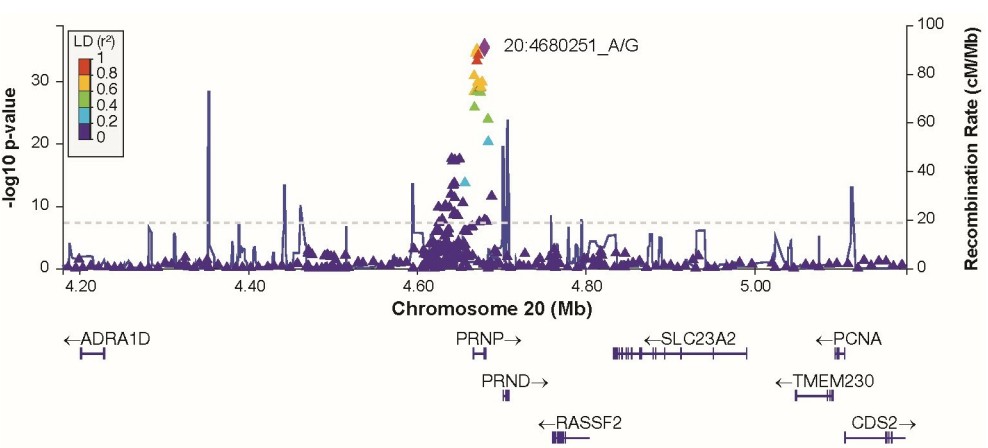

**Fig 6. Regional association plot at *PRNP* locus with clinical duration as phenotype (additive model).**

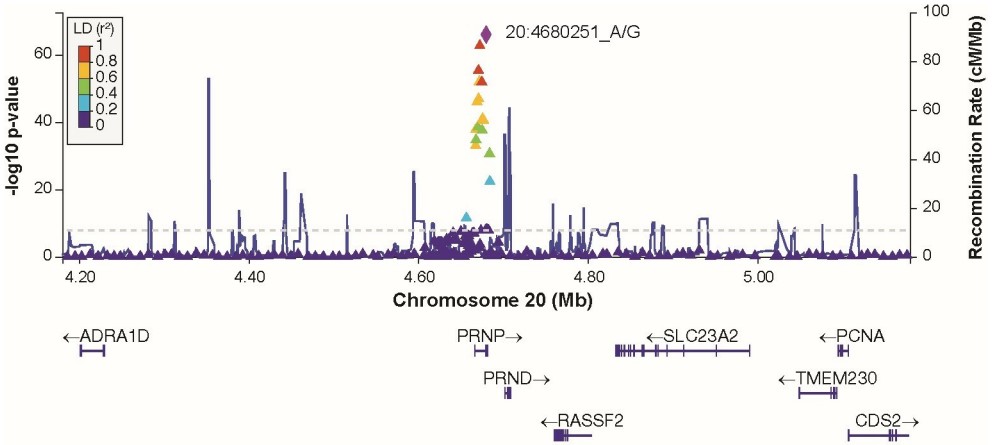

**Fig 7. Regional association plot at *PRNP* locus with clinical duration as phenotype (heterozygous model).**

should be interpreted with caution. Overall, this work further establishes the key importance of the PrP coding sequence relative to other potential mechanisms and genetic loci in determination of CJD survival.

For age at onset there were no genome-wide significant SNPs, but we identified the *HS6ST3* in a gene-based test and intracellular oxygen homeostasis by pathway analysis (S3–S5 Tables). *HS6ST3* or Heparan Sulfate 6-O-Sulfotransferase 3 catalyses the transfer of sulfate from 3'-phosphoadenosine 5'-phosphosulfate (PAPS) to position 6 of the N-sulfoglucosamine residue (GlcNS) of heparan sulfate (HS), thus potentially modifying the interactions of this molecule with cell surface proteins. There is a vast literature on a role for polyanionic compounds, including HS in prion disease pathogenesis, as they colocalise with PrP$^C$ on the cell surface and with aggregated PrP$^{Sc}$ [29], act as potential co-factors in prion replication, and there is potent inhibitory activity of HS and related compounds on prion propagation [30]. A role for intracellular oxygen homeostasis is less clearly linked to prion disease. Both associations were borderline in significance taking into account multiple testing. We found no evidence of genetic correlation between the case-only and published case-control GWAS analyses. We observed only a moderate heritability (h$^2_{SNP}$ = 0.18–0·26, using different methods) for the

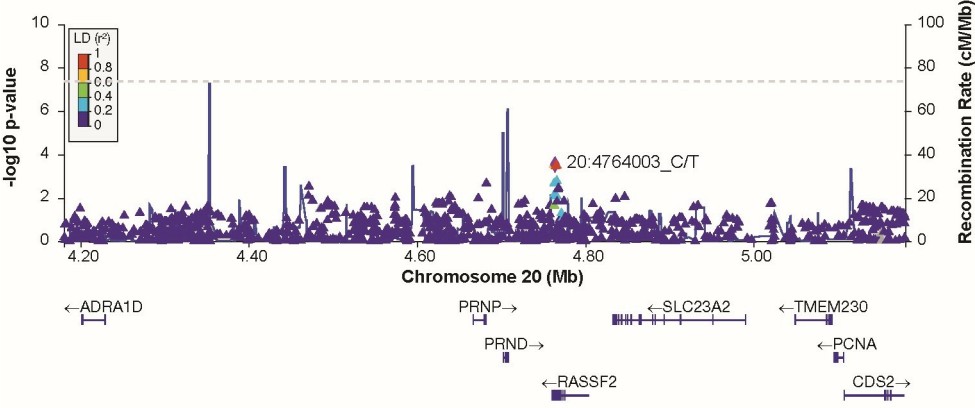

**Fig 8. Regional association plot at *PRNP* locus for conditional analysis on SNP rs1799990 with clinical duration as phenotype (additive model).**

**Table 2. Top 10 genes identified by MAGMA (standalone) gene analysis (including genome-wide significant SNPs) with duration as phenotype.**

| Gene | NCBI Gene ID | Chr | Start (hg19) | Stop (hg19) | NSNPS | N | ZSTAT | Pvalue | Bonf. corr. Pvalue |
|------|-------------|-----|--------------|-------------|-------|---|-------|--------|---------------------|
| PRNP | 5621 | 20 | 4641797 | 4707235 | 189 | 3773 | 6.11 | $5.00 \times 10^{-10}$ | $9.02 \times 10^{-6}$ |
| ANP32E | 81611 | 1 | 150165717 | 150233504 | 82 | 3773 | 4.15 | $1.64 \times 10^{-5}$ | 0.31 |
| CA14 | 23632 | 1 | 150204554 | 150262478 | 59 | 3773 | 3.92 | $4.38 \times 10^{-5}$ | 0.79 |
| TMEM121B | 27439 | 22 | 17572189 | 17627257 | 148 | 3773 | 3.90 | $4.81 \times 10^{-5}$ | 0.87 |
| HDHD5 | 27440 | 22 | 17593410 | 17671177 | 299 | 3773 | 3.89 | $5.10 \times 10^{-5}$ | 0.92 |
| IL17RA | 23765 | 22 | 17540849 | 17621584 | 185 | 3773 | 3.85 | $5.98 \times 10^{-5}$ | 1 |
| CNTN3 | 5067 | 3 | 74286719 | 74688587 | 845 | 3773 | 3.58 | $1.70 \times 10^{-4}$ | 1 |
| APH1A | 51107 | 1 | 150212799 | 150266609 | 57 | 3773 | 3.54 | $2.03 \times 10^{-4}$ | 1 |
| U2SURP | 23350 | 3 | 142695366 | 142804567 | 170 | 3773 | 3.50 | $2.36 \times 10^{-4}$ | 1 |
| FZD8 | 8325 | 10 | 35902177 | 35955362 | 83 | 3773 | 3.47 | $2.58 \times 10^{-4}$ | 1 |

(NSNPS = number of SNPs annotated to a gene; N = number of samples; ZSTAT = Z-score for the gene, based on its p-value)

case-control GWAS [19], and low heritability for the duration phenotype ($h^2_{SNP}$ = 0.09 using LDSC). Common SNPs measured in these studies therefore explain only a small proportion of disease phenotypes. The only locus common to both GWAS studies is *PRNP*, with no evidence that SNPs at the *STX6* or *GAL3ST1* loci have any effect on clinical phenotypes in lead SNP association, gene-based or pathway analyses. It is possible that larger sample sizes, with additional risk factor discovery, will uncover shared determinants, but the current evidence suggests that beyond *PRNP*, distinct mechanisms and/or stochasticity determines disease risk, age at onset and clinical duration.

Absence of an association between *PRNP* cis-eQTL SNPs and clinical duration/age of onset should not deter the pursuit of methods to reduce PrP as a therapeutic strategy. There is a wealth of evidence for the safety and potential effectiveness of this approach from animal models [31–35]. *PRNP* cis-eQTL SNPs are predominantly associated with localised tissue expression of PrP, typically in cerebellum or cerebellar hemispheres, and are relatively modest effects. Therapeutic strategies aim for more profound protein knock-down, which will be critical to achieve across a wide range of central nervous system tissues and cell types [36].

Poleggi et al. (2018) [37] aimed to identify additional genetic modifiers in a GWAS study with a small cohort of patients (E200K mutation only). In this study, two SNPs were identified

**Table 3. Top 10 genes identified by FUMA gene analysis (including genome-wide significant SNPs) with duration as phenotype.**

| Gene | Chr | Start (hg19) | Stop (hg19) | NSNPS | N | ZSTAT | Pvalue | Bonf. corr. Pvalue |
|------|-----|--------------|-------------|-------|---|-------|--------|---------------------|
| PRNP | 20 | 4666882 | 4682236 | 27 | 3773 | 7.02 | $1.12 \times 10^{-12}$ | $2.03 \times 10^{-8}$ |
| CECR5 | 22 | 17618401 | 17646177 | 81 | 3773 | 4.28 | $9.42 \times 10^{-6}$ | 0.17 |
| ANP32E | 1 | 150190717 | 150208504 | 19 | 3773 | 3.86 | $5.63 \times 10^{-5}$ | 1 |
| CA14 | 1 | 150229554 | 150237478 | 3 | 3773 | 3.79 | $7.60 \times 10^{-5}$ | 1 |
| AL356356.1 | 1 | 150521897 | 150524367 | 1 | 3773 | 3.67 | $1.23 \times 10^{-4}$ | 1 |
| AC006946.15 | 22 | 17602476 | 17612994 | 40 | 3773 | 3.51 | $2.20 \times 10^{-4}$ | 1 |
| CCDC174 | 3 | 14693271 | 14714166 | 66 | 3773 | 3.51 | $2.26 \times 10{-4}$ | 1 |
| KCNJ3 | 2 | 155554811 | 155714863 | 453 | 3773 | 3.45 | $2.82 \times 10^{-4}$ | 1 |
| VRK3 | 19 | 50479724 | 50529203 | 115 | 3773 | 3.38 | $3.56 \times 10^{-4}$ | 1 |
| U2SURP | 3 | 142683339 | 142779567 | 154 | 3773 | 3.38 | $3.61 \times 10^{-4}$ | 1 |

(NSNPS = number of SNPs annotated to a gene; N = number of samples; ZSTAT = Z-score for the gene, based on its p-value)

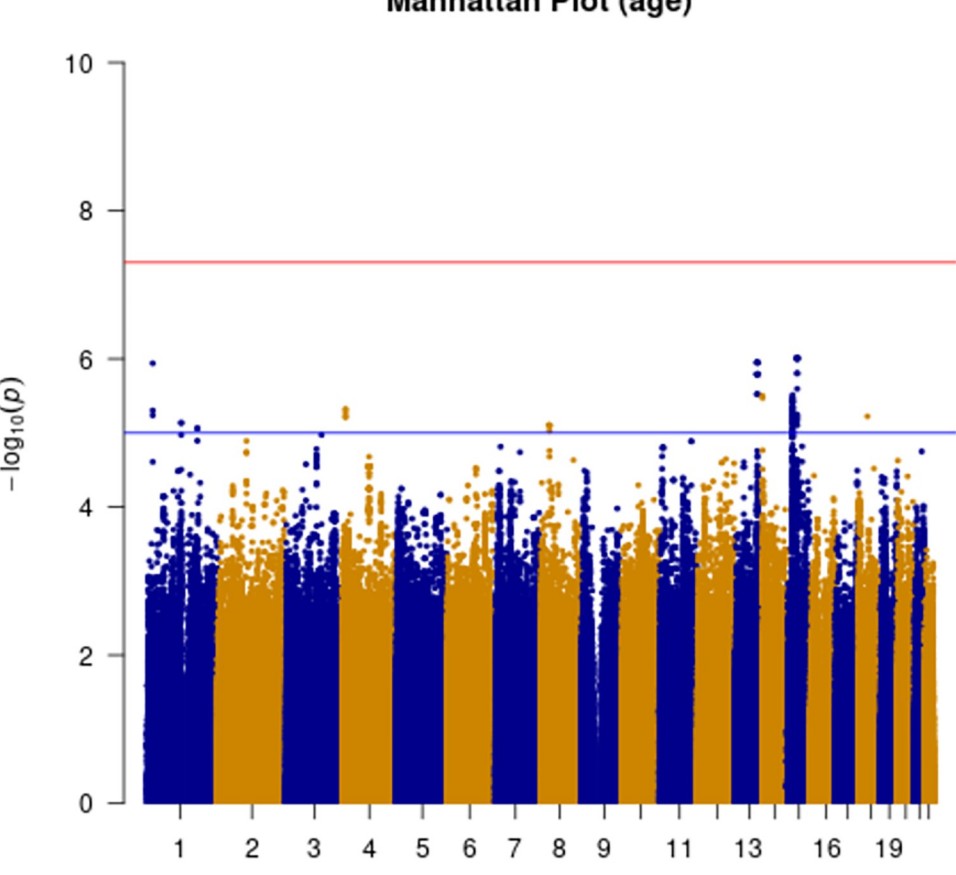

**Fig 9. Manhattan plot with age at onset as phenotype.** (red line indicating genome-wide significance of $5 \times 10^{-8}$; blue line indicating suggestive genome-wide significance ($5 \times 10^{-8} >$ pvalue $< 1 \times 10^{-}$).

within the *CYP4X1* gene locus indicating that this gene modulates onset of disease in sCJD. The top SNP identified in the Poleggi analysis (rs9793471) had a pvalue of 0.08 in our analysis.

A number of GWAS studies reporting genetic modifiers in other neurological diseases of in relation to the age at onset phenotype have been reported. One example is the case-only study of Li et al. [38] where a number of novel genes for age-at onset in Alzheimer's disease were identified. Blauwendraat et al. [39] described several modifier loci in an age-at-onset GWAS analysis of Parkinson's disease.

It was imperative to transform the non-normal distribution of the duration phenotype data as the GWAS association model requires Gaussian distributed phenotype data to avoid model misspecification, which could lead to false conclusions. A number of data transformations were tested (log, rank inverse, square root) for transformation of the phenotype data (duration and age) and the Box-Cox transformation was found to be the best option for establishing the optimal correlation coefficient ensuring a normal distribution and reduction of data noise to a minimum.

This study was limited by sample size and was restricted to the examination of age at onset and clinical duration phenotypes that are almost universally collected, whereas the diversity of clinical phenotypes in CJD is well known (including variable involvement of

**Table 4. Top 10 pathways identified by MAGMA (standalone) gene-set analysis (including genome-wide SNPs) using duration as phenotype.**

| Category | Pathway | NGENES | BETA | Pvalue | Bonf. Corr. Pvalue |
|---|---|---|---|---|---|
| Gene ontology | regulation of calcium ion import across plasma membrane | 2 | 2.86 | $2.57 \times 10^{-6}$ | 0.04 |
| Gene ontology | regulation of T-lymphocyte activation via T cell receptor contact with MHC-bound antigen | 5 | 1.58 | $5.41 \times 10^{-6}$ | 0.09 |
| Gene ontology | cellular response to copper | 11 | 1.04 | $2.55 \times 10^{-5}$ | 0.43 |
| Gene ontology | proteosomal ubiquitin-independent protein catabolic process | 4 | 1.51 | $6.25 \times 10^{-5}$ | 1.00 |
| Gene ontology | anchored component of external side of plasma membrane | 18 | 0.77 | $1.06 \times 10^{-4}$ | 1 |
| Gene ontology | response to iron ion | 30 | 0.61 | $1.11 \times 10^{-4}$ | 1 |
| Gene ontology | obsolete intrinsic component of external side of plasma membrane | 23 | 0.69 | $1.16 \times 10^{-4}$ | 1 |
| Gene ontology | T cell activation via T cell receptor contact with antigen bound to MHC molecule on antigen presenting cell | 8 | 1.06 | $1.27 \times 10^{-4}$ | 1 |
| Gene ontology | CD4-positive, CD25-positive, alpha-beta regulatory T cell differentiation | 4 | 1.33 | $3.44 \times 10^{-4}$ | 1 |
| Gene ontology | positive regulation of T cell activation via T cell receptor contact with antigen bound to MHC molecule on antigen presenting cell | 2 | 1.67 | $3.52 \times 10^{-4}$ | 1 |

(NGENES = number of genes in the gene-set dataset; BETA = regression coefficient of the gene set)

cognitive, ataxic, psychiatric, sleep and motor aspects). In biochemical aspects and biomarkers, we see diversity of $PrP^{Sc}$ types, and different imaging, neurophysiological and fluid biomarker associations. These parameters are only collected in smaller subsets of data. Genetic studies in a rare disease like sCJD benefit from national investment and collaboration in prion disease surveillance [40]. Future work of the collaborative group might focus on building larger sample collections for increased power, exome or genome studies to ascertain rare and structural variants and extension of these type of analyses to other phenotypes (e.g., the well-known subtypes of CJD based on major symptom at presentation (ataxia, visual processing disorder etc.)).

**Table 5. Top 10 pathways identified by FUMA gene-set analysis (including genome-wide SNPs) using duration as phenotype.**

| Category | Pathway | NGENES | BETA | Pvalue | Bonf. corr. Pvalue |
|---|---|---|---|---|---|
| Gene ontology | type_5_metabotropic_glutamate_receptor_binding | 5 | 1.92 | $1.85 \times 10^{-5}$ | 0.29 |
| Gene ontology | ureteric_bud_elongation | 9 | 1.00 | $8.58 \times 10^{-5}$ | 1 |
| Gene ontology | negative_regulation_of_cell_maturation | 8 | 0.89 | $1.53 \times 10^{-5}$ | 1 |
| Gene ontology | mechanosensory_behavior | 13 | 0.85 | $1.55 \times 10^{-5}$ | 1 |
| Gene ontology | actin_filament_based_transport | 8 | 0.90 | $3.84 \times 10^{-5}$ | 1 |
| Gene ontology | learned_vocalization_behavior_or_vocal_learning | 8 | 0.97 | $3.99 \times 10^{-5}$ | 1 |
| Gene ontology | peptidyltransferase_activity | 3 | 1.70 | $5.78 \times 10^{-5}$ | 1 |
| Gene ontology | pyrimidine_containing_compound_transmembrane_transport | 10 | 0.79 | $7.38 \times 10^{-5}$ | 1 |
| Curated gene sets | smid_breast_cancer_relapse_in_pleura_dn | 24 | 0.49 | $7.89 \times 10^{-5}$ | 1 |
| Gene ontology | vitamin_binding | 128 | 0.23 | $8.75 \times 10^{-4}$ | 1 |

(NGENES = number of genes in the gene-set dataset; BETA = regression coefficient of the gene set)

## Supporting information

**S1 Text. Patient recruitment and phenotypes.**
(DOCX)

**S1 Fig. Box-Cox normality plot showing the correlation coefficient (maximum value at -0.14).**
(TIF)

**S2 Fig. Principal component analysis with first two axes used for exclusion of case and control samples (1000 Genome data used as European ancestry control).**
(TIF)

**S3 Fig. Principal component featuring distribution of samples in terms of country of origin.**
(TIF)

**S4 Fig. Regional association plot at *PRNP* locus for conditional analysis on SNP rs1799990 with duration as phenotype (heterozygous model).**
(TIF)

**S5 Fig. Regional association plot at suggestive locus (*HDHD5*) with duration as phenotype (additive model).**
(TIF)

**S6 Fig. Regional association plot at suggestive *FHIT* locus with duration as phenotype (additive model).**
(TIF)

**S7 Fig. Regional association plot at suggestive *EREG* locus with duration as phenotype (additive model).**
(TIF)

**S8 Fig. Regional association plot at suggestive *NEDD4* locus with age as phenotype (additive model).**
(TIF)

**S9 Fig. Regional association plot at suggestive *UGGT2* locus with age as phenotype (additive model).**
(TIF)

**S10 Fig. Meta-analysis of case-only and case-control GWAS.**
(TIF)

**S11 Fig. Power analysis indicating the strength of power of *PRNP* lead SNP (rs1799990) and lead SNPs of suggestive hits as described in the Results section based on a range of effect sizes vs. MAF.**
(TIF)

**S12 Fig. Histograms for phenotype age before and after Box-Cox transformation for codon 129 genotypes MM, MV and VV.**
(TIF)

**S13 Fig. Histograms for phenotype duration before and after Box-Cox transformation for codon 129 genotypes MM, MV and VV.**
(TIF)

**S1 Table. List of 53 significantly associated SNPs (Pvalue $< 5 \times 10^{-8}$).**
(XLSX)

**S2 Table. List of 51 suggestively associated SNPs ($5 \times 10^{-8} >$ Pvalue $< 1 \times 10^{-5}$).**
(XLSX)

**S3 Table. Top 10 genes identified by MAGMA (standalone) gene analysis (including genome-wide significant SNPs) with age as phenotype.**
(XLSX)

**S4 Table. Top 10 genes identified by FUMA gene analysis (including genome-wide significant SNPs) with age as phenotype.**
(XLSX)

**S5 Table. Top 10 pathways identified by MAGMA (standalone) gene-set analysis (including genome-wide SNPs) using age as phenotype.**
(XLSX)

**S6 Table. Genetic correlation using LDSC with a genetic correlation of 0.1467.**
(XLSX)

**S7 Table. Top 10 genes identified by MAGMA involved in sphingolipid and intracellular trafficking pathways (including genome-wide significant SNPs) with age as phenotype.** Green rectangles indicate genes involved in neurological diseases (PrD = Prion Disease, PD = Parkinson's Disease, AD = Alzheimer's Disease, ALS = Amyotrophic Lateral Sclerosis; HD = Huntington's Disease, FTD = Frontotemporal Dementia) (NSNPS = number of SNPs annotated to a gene; N = number of samples; ZSTAT = Z-score for the gene, based on its p-value).
(XLSX)

**S8 Table. Top 10 genes identified by MAGMA involved in sphingolipid and intracellular trafficking pathways (including genome-wide significant SNPs) with duration as phenotype.** Green rectangles indicate genes involved in neurological diseases (PrD = Prion Disease, PD = Parkinson's Disease, AD = Alzheimer's Disease, ALS = Amyotrophic Lateral Sclerosis; HD = Huntington's Disease, FTD = Frontotemporal Dementia). (NSNPS = number of SNPs annotated to a gene; N = number of samples; ZSTAT = Z-score for the gene, based on its p-value).
(XLSX)

**S9 Table. Top 10 pathways identified by FUMA gene-set analysis (including genome-wide SNPs) using age as phenotype. (NGENES = number of genes in the gene-set dataset; BETA = regression coefficient of the gene set).**
(XLSX)

## Acknowledgments

We thank Richard Newton for support with images and UCL Genomics who did the array processing. For UK samples we would like to thank patients, their families and carers, UK neurologists and other referring physicians, co-workers at the National Prion Clinic, our colleagues at the National Creutzfeldt-Jakob Disease Research and Surveillance Unit, Edinburgh. We thank Dr. Maria Styczynska from Mossakowski Medical Research Centre; Polish Academy of Sciences; Warsaw, for kindly providing control DNA samples for the Polish cohort. We thank Inés Santiuste and the Valdecilla Biobank (PT17/0015/0019), integrated in the Spanish

Biobank Network, for their support and collaboration in sample collection and management. We thank Megan Casey for assistance with sample collection and management. The Australian National Creutzfeldt-Jakob Disease Registry (ANCJDR) would like to thank all patients and their families for supporting surveillance activities that have allowed participation in the study, as well as their managing physicians. The Italian Creutzfeldt-Jakob Registry would like to thank Dr.Clara Salciccia for the collaboration in DNA sample collection and management, Cinzia Gasparrini for administrative assistance and all patients, their families, neurologists, and referring physicians.

## Author Contributions

**Formal analysis:** Holger Hummerich.

**Investigation:** Helen Speedy, Tracy Campbell, Lee Darwent, Elizabeth Hill.

**Resources:** Steven Collins, Christiane Stehmann, Gabor G. Kovacs, Michael D. Geschwind, Karl Frontzek, Herbert Budka, Ellen Gelpi, Adriano Aguzzi, Sven J. van der Lee, Cornelia M. van Duijn, Pawel P. Liberski, Miguel Calero, Pascual Sanchez-Juan, Elodie Bouaziz-Amar, Jean-Louis Laplanche, Stéphane Haïk, Jean-Phillipe Brandel, Angela Mammana, Sabina Capellari, Anna Poleggi, Anna Ladogana, Maurizio Pocchiari, Saima Zafar, Stephanie Booth, Gerard H. Jansen, Aušrinė Areškevičiūtė, Eva Løbner Lund, Katie Glisic, Piero Parchi, Peter Hermann, Inga Zerr, Brian S. Appleby, Jiri Safar, Pierluigi Gambetti.

**Supervision:** John Collinge, Simon Mead.

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
