## [Decision Letter · Decision Letter 0]

12 Mar 2024

PONE-D-24-01351Genome wide association study of clinical duration and age at onset of sporadic CJDPLOS ONE

Dear Dr. Hummerich,

Thank you for submitting your manuscript to PLOS ONE. After careful consideration, we feel that it has merit but does not fully meet PLOS ONE’s publication criteria as it currently stands. Therefore, we invite you to submit a revised version of the manuscript that addresses the points raised during the review process.

Please, provide changes according to reviewers' criticisms reported below.

We look forward to receiving your revised manuscript.

Kind regards,

Gianluigi Zanusso

Academic Editor

PLOS ONE

“Dr. Budka reports grants from Federal Office for Health, Swiss Government, during the conduct of the study. Dr. HAIK reports grants from Santé Publique France, during the conduct of the study; grants from LFB Biomedicaments, grants from Institut de Recherche Servier, grants from MedDay Pharmaceuticals, outside the submitted work; In addition, Dr. HAIK has a patent Method for treating prion diseases (PCT/EP2019/070457) pending. Dr. Appleby reports grants from Centers for Disease Control and Prevention, during the conduct of the study. Fronztek reports grants from Ono Pharmaceuticals outside the submitted work. Dr. Mead reports grants from Medical Research Council (UK) and grants from National Institute of Health Research’s Biomedical Research Centre at University College London Hospitals NHS Foundation Trust during the conduct of the study. Dr. Kovacs reports personal fees from Biogen, outside the submitted work. Dr. Collinge reports grants from Medical Research Council, grants from NIHR UCLH Biomedical Research Centre, during the conduct of the study; and is a Director and shareholder of D-Gen Limited, an academic spinout in the field of prion disease diagnostics, decontamination and therapeutics. Dr. Pocchiari reports personal fees from Ferring Pharmaceuticals, personal fees from CNCCS (Collection of National Chemical Compounds and Screening Center), non-financial support from Fondazione Cellule Staminali, outside the submitted work. Dr Geschwind has consulted for3D Communications, Adept Field Consulting, Advanced Medical Inc., Best Doctors Inc., Second Opinion Inc., Gerson Lehrman Group Inc., Guidepoint Global LLC, InThought Consulting Inc., Market Plus, Trinity Partners LLC, Biohaven Pharmaceuticals, Quest Diagnostics and various medical-legal consulting. He has received speaking honoraria for various medical center lectures and from Oakstone publishing. He has received past research support from Alliance Biosecure, CurePSP, the Tau Consortium, and Quest Diagnostics. Dr. Geschwind serves on the board of directors for San Francisco Bay Area Physicians for Social Responsibility and on the editorial board of Dementia & Neuropsychologia.”

5. We notice that your supplementary figures and tables are included in the manuscript file. Please remove them and upload them with the file type 'Supporting Information'. Please ensure that each Supporting Information file has a legend listed in the manuscript after the references list.

Reviewers' comments:

**Comments to the Author**

1. Is the manuscript technically sound, and do the data support the conclusions?

Reviewer #1: Yes

Reviewer #2: Yes

2. Has the statistical analysis been performed appropriately and rigorously? 

Reviewer #1: Yes

Reviewer #2: Yes

3. Have the authors made all data underlying the findings in their manuscript fully available?

Reviewer #1: Yes

Reviewer #2: Yes

4. Is the manuscript presented in an intelligible fashion and written in standard English?

Reviewer #1: Yes

Reviewer #2: Yes

5. Review Comments to the Author

Reviewer #1: Previous studies identified various factors influencing survival time in sporadic Creutzfeldt-Jakob disease (sCJD), including demographics, prion protein genotype, and biomarkers. This study aimed to establish associations between common genetic variations and key clinical features of sCJD to enhance understanding of disease processes and identify potential therapeutic targets. The authors report the analysis of a large cohort of genetic determinants of clinical duration and age at onset in 3999 sCJD cases derived from previous literature or newly genotyped on Illumina’s Global Screening Array. They identified 53 SNPs on chromosome 20 that were significantly associated with clinical duration. The non-synonymous variant at codon 129 in the PRNP gene emerged as a key determinant of clinical duration. No genome-wide significant SNP determinants were found for age at onset, but the HS6ST3 gene showed significance in a gene-based test. The manuscript confirms that the PRNP codon 129 is essentially the primary modifier of CJD survival, suggesting limited effects at other genetic loci.

The manuscript is well written. The data are analysed robustly and discussed appropriately. The conclusions are adherent with the results.

I have no issues with this manuscript.

Reviewer #2: The paper illustrates the genetic influence, determined by GWA analysis, on CJD phenotypes, clinical duration and age at onset. Although the number of subjects is important, with the exception of polymorphism PRNP codon 129 no other genetic variance affecting the CJD survival.

The experimentation is well conducted and the negative result convincing .

The discussion should be extended in relation to the previous work published by the same consortium (Jones et al Lancet Neurol, 2020) where variants in two loci (STX6 and Gal3ST1) were identified as risk factors in sporadic CJD, for instance in my opinion it is difficult to consider the possibility that an investigation in a larger sample size (line 317) can give different results. Another paper (Poleggi et al J Neurol Neurosurg Psych., 2018) indicating CYP4X1 gene as modulator of sporadic CJD should be considered in the discussion. Positive examples of the approach described in the paper in other neurodegenerative disorders should be also mentioned in the discussion.

6. PLOS authors have the option to publish the peer review history of their article (what does this mean?). If published, this will include your full peer review and any attached files.

Reviewer #1: **Yes: **Emiliano Biasini

Reviewer #2: **Yes: **Gianluigi Forloni

---

## [Author Response · Author response to Decision Letter 0]

26 Apr 2024

Response to reviewers:

We thank Reviewer #1 for his positive comments.

We also thank Reviewer #2 for his helpful and constructive feedback.

In relation to the comment that a larger sample size is unlikely to result in the identification of potential modifiers, we would like to state that we identified SNPs very close to the genome wide significance level of 5*10-8 in gene HDHD5. We have and are currently collecting additional samples and therefore we do not want to rule out the possibility that the addition of these samples might confirm HDHD5 as a risk locus for CJD in the future.

The second comment relates to the publication by Poleggi et al. 2018 identifying CYP4X1 as a candidate disease modifier for sCJD. We have addressed this in the Discussion section of the manuscript 

We also added examples of other GWAS studies of other neurological diseases with age-at-onset phenotype to the Discussion section and cited two papers, one relating to Alzheimer's disease and the other to Parkinson's disease.

---

## [Editor Report · Decision Letter 1]

14 May 2024

Genome wide association study of clinical duration and age at onset of sporadic CJD

PONE-D-24-01351R1

Dear Dr. Mead,

We’re pleased to inform you that your manuscript has been judged scientifically suitable for publication and will be formally accepted for publication once it meets all outstanding technical requirements.

Kind regards,

Gianluigi Zanusso

Academic Editor

PLOS ONE

---

## [Editor Report · Acceptance letter]

17 Jul 2024

PONE-D-24-01351R1 

PLOS ONE

Dear Dr. Hummerich, 

I'm pleased to inform you that your manuscript has been deemed suitable for publication in PLOS ONE. Congratulations! Your manuscript is now being handed over to our production team.

Kind regards, 

on behalf of

Dr. Gianluigi Zanusso 

Academic Editor

PLOS ONE